# Homoplasy in the evolution of modern human-like joint proportions in *Australopithecus afarensis*

**Anjali M Prabhat[1], Catherine K Miller[1,2], Thomas Cody Prang[3], Jeffrey Spear[4,5], Scott A Williams[4,5], Jeremy M DeSilva[1,2]\***

[1]Anthropology, Dartmouth College, Hanover, United States; [2]Ecology, Evolution, Ecosystems, and Society, Dartmouth College, Hanover, United States; [3]Department of Anthropology, Texas A&M University, College Station, United States; [4]Center for the Study of Human Origins, Department of Anthropology, New York University, New York, United States; [5]New York Consortium in Evolutionary Primatology, New York, United States

**Abstract** The evolution of bipedalism and reduced reliance on arboreality in hominins resulted in larger lower limb joints relative to the joints of the upper limb. The pattern and timing of this transition, however, remains unresolved. Here, we find the limb joint proportions of *Australopithecus afarensis*, *Homo erectus*, and *Homo naledi* to resemble those of modern humans, whereas those of *A. africanus*, *Australopithecus sediba*, *Paranthropus robustus*, *Paranthropus boisei*, *Homo habilis*, and *Homo floresiensis* are more ape-like. The homology of limb joint proportions in *A. afarensis* and modern humans can only be explained by a series of evolutionary reversals irrespective of differing phylogenetic hypotheses. Thus, the independent evolution of modern human-like limb joint proportions in *A. afarensis* is a more parsimonious explanation. Overall, these results support an emerging perspective in hominin paleobiology that *A. afarensis* was the most terrestrially adapted australopith despite the importance of arboreality throughout much of early hominin evolution.

**\*For correspondence:**
Jeremy.M.DeSilva@dartmouth.
edu

**Competing interest:** The authors declare that no competing interests exist.

## Introduction

Among extant hominoids, modern humans (*Homo sapiens*; hereafter, 'humans') are the only habitually bipedal species. Adaptation to upright walking and running in humans is evidenced by the presence of a host of postcranial morphologies functionally related to saving mechanical and metabolic energy (*Lovejoy, 1988*; *Bramble and Lieberman, 2004*; *Pontzer, 2017*). These include relatively long legs, arched feet (*Venkadesan et al., 2020*), and adaptations to protect the joints of the lower limbs from excessive stress by increasing their surface areas relative to the mass of the body (*Ruff, 1988*; *Jungers, 1988*; *Lovejoy, 2005*). These morphological traits are most strongly expressed in recent modern humans, which are nearly exclusively terrestrial in their locomotor adaptation. In contrast, the body plan of extant, nonhuman apes (hereafter, simply 'apes') reflects an adaptation to orthogrady and suspension (*Keith, 1923*; *Gebo, 1996*; *Williams and Russo, 2015*) with relatively long arms and large upper limb joints (*Ruff, 1988*), elongated, curved phalanges (*Deane and Begun, 2008*), and other morphological features suitable for arboreal behaviors (*Gebo, 1996*). Although chimpanzees, bonobos, and gorillas possess adaptations to terrestrial quadrupedalism (*Gebo, 1996*), including their knuckle-walking hand posture and heel-strike plantigrade foot posture (*Gebo, 1992*; *Prang, 2019*), they retain traits linked to an ancestry characterized by vertical climbing and suspension in some form (*Gebo, 1996*).

The relatively larger upper limb joints of apes compared to humans reflect disparate joint loading regimes associated with forms of quadrupedalism, climbing, and suspension (*Gebo, 1996*). Additionally, the relatively larger surface areas of the convex side of conarticular joints may contribute to increased range of motion, providing benefits to the arboreal locomotor performance of apes (*Ruff, 1988*; *Godfrey et al., 1991*; *Hammond, 2014*; *Prang, 2016*). Therefore, the relative size of postcranial joints and the relationship between the joints of the upper and lower limbs are important correlates of positional and locomotor behavior among hominoids (*Ruff, 1988*; *Jungers, 1988*; *Godfrey et al., 1995*; *McHenry, 1992*; *McHenry and Berger, 1998*; *Green et al., 2007*; *Haeusler and McHenry, 2007*).

The timing and pattern of the complicated, nonlinear evolutionary loss of adaptations to arboreality and the transition to a form of nearly exclusive terrestrial bipedalism among hominins has been debated for decades (*Stern, 2000*; *Ward, 2002*). The study of limb joint proportions initially focused on the preserved partial skeletons A.L. 288-1 (*Australopithecus afarensis*) and StW 431 (*Australopithecus africanus*) (*McHenry and Berger, 1998*; *Green et al., 2007*), along with OH 62 and KNM-ER 3735 (*Homo habilis*; *Haeusler and McHenry, 2007*; *Johanson et al., 1987*; *Leakey et al., 1987*). Previous studies have shown that the geologically younger *A. africanus* possessed relatively large upper limb joints and metaphyseal dimensions in comparison to *A. afarensis* (*McHenry and Berger, 1998*; *Green et al., 2007*). The OH 62 and KNM-ER 3735 partial skeletons are more fragmentary, but morphological comparisons of external morphology (*Haeusler and McHenry, 2007*; *Hartwig-Scherer and Martin, 1991*) and cross-sectional geometry (*Ruff, 2009*) suggest that the upper limbs of *H. habilis* bore similarities to extant chimpanzees and gorillas, implying greater reliance on forelimb-dominated behaviors, and may show a similar pattern to that observed in *A. africanus* (*McHenry and Berger, 1998*; *Green et al., 2007*; *Haeusler and McHenry, 2007*). The observed pattern of joint size proportions among extant hominoids implies that the relatively larger lower limb joints of *A. afarensis* are a reflection of increased terrestriality compared to *A. africanus*.

Cladistic analyses of hominin phylogeny based on craniodental characters consistently position *A. africanus* as more closely related to *Homo* than is *A. afarensis* (*Dembo et al., 2016*; *Strait et al., 2015*). Therefore, either (1) *A. afarensis* and *H. sapiens* independently evolved relatively larger lower limb joints (i.e., their similarities are homoplastic), (2) *A. africanus* and *H. habilis* evolved more ape-like joint proportions from an ancestor with more human-like limb proportions (i.e., the similarities between *A. africanus*, *H. habilis*, and apes are homoplastic), or (3) *A. afarensis* is more closely related to *Homo* than are *A. africanus* and *H. habilis* (*Figure 1*). Limited taxonomic sampling of fossil hominins in previous studies has rendered these competing scenarios exceedingly difficult to differentiate. Over the past few decades, however, the recovery of new fossil hominin partial skeletons preserving both upper and lower limb joints has provided an expanded sample that can be used to evaluate these hypotheses more rigorously (*Table 1*). Here, we re-examine the upper and lower limb joint proportions of multiple species of *Australopithecus*, *Paranthropus*, and *Homo* to evaluate these long-standing alternative hypotheses for patterns of postcranial evolution in hominins.

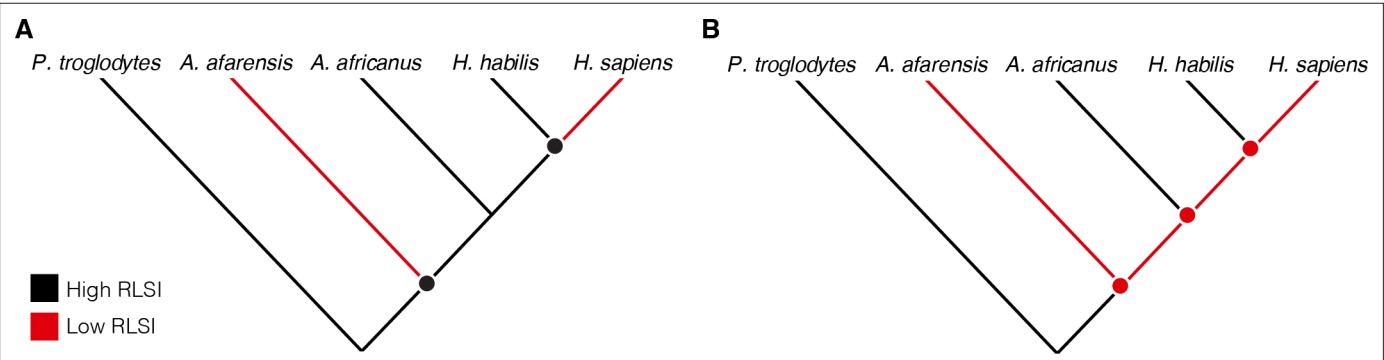

**Figure 1.** Alternative hypotheses to explain the pattern of limb joint proportions observed in the human fossil record. Previous work interpreted the human-like ratio of upper to lower limb joint size (relative limb size index [RLSI]) in *Australopithecus afarensis* to indicate either (**A**) homoplasy between *A. afarensis* and *Homo sapiens* or (**B**) evolutionary reversals to a more ape-like body form in *A. africanus* and *H. habilis*.

**Table 1.** Fossil hominin and extant hominoid measurements.

| Specimen | Taxon | G | H | B | U | R | F | Sub | A | T | Sac |
|---|---|---|---|---|---|---|---|---|---|---|---|
| | Homo sapiens | 29.5 ± 2.7 (N = 67) | 42.3 ± 3.5 (N = 67) | 59.1 ± 4.7 (N = 52) | 21.8 ± 1.9 (N = 51) | 21.7 ± 2.1 (N = 51) | 44.5 ± 3.5 (N = 67) | 28.6 ± 2.1 (N = 52) | 51.2 ± 3.5 (N = 67) | 28.2 ± 2.0 (N = 66) | 38.6 ± 3.1 (N = 67) |
| | Pan | 26.6 ± 2.4 (N = 113) | 38.2 ± 3.1 (N = 113) | 62.5 ± 5.5 (N = 95) | 22.5 ± 2.8 (N = 95) | 24.5 ± 1.8 (N = 94) | 32.8 ± 2.5 (N = 120) | 25.0 ± 2.0 (N = 98) | 38.6 ± 3.3 (N = 116) | 18.1 ± 34.1 (N = 116) | 28.4 ± 3.9 (N = 109) |
| | Gorilla | 39.2 ± 5.5 (N = 119) | 54.9 ± 7.3 (N = 122) | 93.0 ± 13.0 (N = 94) | 33.4 ± 5.7 (N = 89) | 31.7 ± 4.4 (N = 91) | 46.6 ± 5.9 (N = 125) | 35.7 ± 4.9 (N = 93) | 53.1 ± 6.9 (N = 114) | 24.9 ± 5.6 (N = 108) | 37.7 ± 5.8 (N = 102) |
| | Pongo | 29.2 ± 3.6 (N = 47) | 40.0 ± 4.8 (N = 49) | 63.9 ± 7.1 (N = 45) | 22.1 ± 3.2 (N = 46) | 22.8 ± 2.8 (N = 46) | 32.9 ± 4.0 (N = 49) | 20.8 ± 2.6 (N = 45) | 39.0 ± 4.7 (N = 49) | 18.0 ± 2.7 (N = 46) | 28.0 ± 4.1 (N = 43) |
| | Hylobatids | 13.2 ± 1.7 (N = 62) | 18.5 ± 2.3 (N = 66) | 28.0 ± 3.1 (N = 66) | 11.0 ± 1.5 (N = 66) | 12.6 ± 1.5 (N = 69) | 16.4 ± 2.1 (N = 65) | 11.0 ± 1.5 (N = 65) | 20.6 ± 3.0 (N = 66) | 7.5 ± 1.0 (N = 59) | 14.7 ± 2.3 (N = 58) |
| A.L. 288-1 | A. afarensis | 21.6 | 28.9 | 41.1 | 16.1 | 15.1 | 28.6 | 20.8 | 37.0 | 18.0 | 25.3 |
| KSD-VP-1/1 | A. afarensis | 30.1 | | 58.8 | | | | | 49.0 | | 32.4 |
| DIK-1-1 | A. afarensis | 13.5 | | | | | | | | 13.1 | |
| StW 573 | A. prometheus (?); A. africanus | 25.9 | 31.3 | 54.0 | 24.3 | 21.9 | 35.2 | 24.5 | 43.0 | 18.0 | |
| StW 431 | A. africanus | | | 59.0 | 25.7 | 21.9 | | | 45.0 | | 27.5 |
| MH1 | A. sediba | | | 57.0 | 18.9 | | 33.0 | 23.2 | | | 22.0 |
| MH2 | A. sediba | 24.6 | 30.1 | 52.4 | 17.4 | 18.8 | 32.7 | | | 18.1 | 23.6 |
| BOU-VP-12/1 | A. garhi(?) | | | | | 21.4 | | 23.7 | | | |
| TM 1517 | P. robustus | | | 54.0 | 22.0 | | | | | 18.9 | |
| OH 80 | P. boisei | | | | | 26.3 | | 26.5 | | | |
| KNM-ER 1500 | P. boisei | | | | | 21.4 | 20.2 | 24.2 | | 19.2* | |
| KNM-ER 1503/1504 | P. boisei | | | 57.0 | | | | 30.6 | 22.2 | | |
| KNM-ER 3735 | H. habilis | | | 55.0 | | 20.0 | | | | | 25.3 |
| KNM-WT 15000 | H. erectus | 27.6 | 31.6 | 55.0 | 19.0 | | 46.0 | 28.8 | | 25.0* | 33.6 |
| LES 1 | H. naledi | | 33.2 | | 16.1 | | 36.0 | 24.2 | | | 24.5 |
| LB 1 | H. floresiensis | | | | 19.5 | | 31.0 | 22.1 | 36.0 | 19.5 | |

*Estimated from tibial plafond width.

G: glenoid size (geomean of SI height and AP width); H: humeral head diameter (geomean of SI height and AP width); B: humeral biepicondylar breadth; U: ulna olecranon width; R: radial head diameter; F: femoral head diameter; Sub: femoral subtrochanteric width (geomean of ML width and AP breadth); A: acetabulum height; T: talar mediolateral width; Sac: sacral size (geomean of ML width and AP breadth). ±1 standard deviation is given with sample size (N=#).

## Results

We quantified limb joint proportions per individual using the relative limb size index (RLSI) (*Green et al., 2007*). The RLSI is the logged ratio of geometric means calculated from upper (forelimb) and lower (hindlimb) limb measurements and quantifies whether a given specimen has relatively larger forelimb or hindlimb joints (*Green et al., 2007*). We calculated a series of RLSIs to accommodate the differential preservation of postcranial elements among the 16 hominin partial skeletons sampled here. When the full upper to lower limb dataset is used, there is clear separation between humans, with their proportionally larger lower limbs, and modern apes, with their proportionately larger upper limbs, with no overlap. Importantly, there remains clear separation between humans and great apes in cases where truncated datasets were used to quantify the limb joint proportions of less complete

hominin skeletons. The ape data, however, do not always accord with degree of arboreality (hylobatid > *Pongo* > *Pan* > *Gorilla*).

5 of the 16 partial hominin skeletons are human-like in their limb joint proportions (*Figure 2*, *Figure 2—figure supplements 1–26*). The RLSI of A.L. 288-1 (Lucy) is far outside the ape range and falls squarely within the range of modern humans (*Figure 2*, *Figure 2—figure supplement 1*). Likewise, the larger, presumed male *A. afarensis* partial skeleton KSD-VP-1/1 is positioned within the human range, though it overlaps with the low end of the hylobatid distribution (*Figure 2—figure supplement 2*). The infant partial skeleton of *A. afarensis* (DIK-1-1), as well as Lucy, has a human-like glenoid:talus ratio (*Figure 2—figure supplement 3*). KNM-WT 15000 (*Homo erectus*) has even larger relative lower limb joint proportions than the humans sampled in this study and is well outside the ape range (*Figure 2—figure supplement 4*). LES 1 (*Homo naledi*) falls within the human interquartile range, outside any modern ape distribution (*Figure 2—figure supplement 5*).

All of the other hominin skeletons studied fall outside the human range, indicating that they are more ape-like in their joint proportions (*Figure 2*). StW 431 (*A. africanus*) has limb proportions positioned within the hylobatid interquartile range and within the distributions of *Pan*, *Gorilla*, and *Pongo* (*Figure 2—figure supplement 6*). MH1 (juvenile *Australopithecus sediba*) falls within the interquartile range of hylobatids and *Pan* and within the ranges of *Gorilla* and *Pongo* (*Figure 2—figure supplement 7*). MH2 (adult *A. sediba*) occupies the space between great apes and humans, positioned only within the range of hylobatids (*Figure 2—figure supplement 8*). StW 573 is similar to MH2 in having relatively larger upper limb joints than modern humans but smaller than extant apes, positioned only near a hylobatid outlier (*Figure 2—figure supplement 9*).

Partial skeletons attributed to *Paranthropus* all possess ape-like joint proportions. TM 1517 (*Paranthropus robustus*) and KNM-ER 1500 (*Paranthropus boisei*) fall within the ranges of all four extant apes (*Figure 2—figure supplements 10 and 11*). OH 80 (*P. boisei*) falls within the interquartile range of *Pan* and the range of *Gorilla* and hylobatids (*Figure 2—figure supplement 12*). Associated fossils KNM-ER 1503/1504 (tentatively attributed to *P. boisei*) have joint proportions within the interquartile range of *Pan*, *Gorilla*, and hylobatids (*Figure 2—figure supplement 13*).

BOU-VP-12/1 (*Australopithecus* cf. *garhi*) falls squarely within the *Gorilla* interquartile range and within the lower range of *Pan* (*Figure 2—figure supplement 14*). There is a single human outlier overlapping with the limb joint proportions of BOU-VP-12/1.

KNM-ER 3735 (*H. habilis*) has joint proportions in the hylobatid interquartile range and within the ranges of all extant great apes (*Figure 2—figure supplement 15*). The joint proportions of LB 1 (*Homo floresiensis*) fall within the ranges for all of the apes, though there is a single human outlier similar in joint proportions to LB 1 (*Figure 2—figure supplement 16*).

Parsimony reconstructions suggest that the human-like limb joint proportions in *A. afarensis* and modern humans are homoplastic, regardless of the phylogenetic hypothesis used (*Figure 3*). The only phylogenetic hypothesis that does not require either homoplasy or multiple reversals is one in which *A. afarensis* is more derived than *H. habilis*, *H. floresiensis*, and all other species of *Australopithecus* and *Paranthropus* examined in this study. Given that an *A. afarensis*-later *Homo* clade (to the exclusion of early *Homo*) has not been supported by any phylogenetic analysis, we consider this last scenario unlikely in the extreme.

## Discussion

Apes have relatively larger upper limb than lower limb joints as reflected by their higher RLSI than modern humans (*Green et al., 2007*). With musculoskeletal anatomies adapted for climbing and suspension, apes possess larger upper limb muscles and joints with greater surface areas, which has the effect of limiting excessive stresses and strains arising from large joint reaction forces. Enlarged upper limb joint surface areas in apes may also contribute to increased ranges of motion (e.g., at the glenohumeral joint). In contrast, humans are characterized by relatively larger lower limb joints, which act to reduce stresses and strains on the joints and nonrenewable cartilage of the hip, knee, and ankle arising from repetitive high-magnitude ground and joint reaction forces during heel-striking bipedal walking and running. This pattern accords with expectations based on the posture and locomotion of apes and humans. Apes possess heavily built upper limbs associated with orthograde climbing and suspension, whereas modern humans have robust lower limbs adapted to terrestrial bipedalism. This

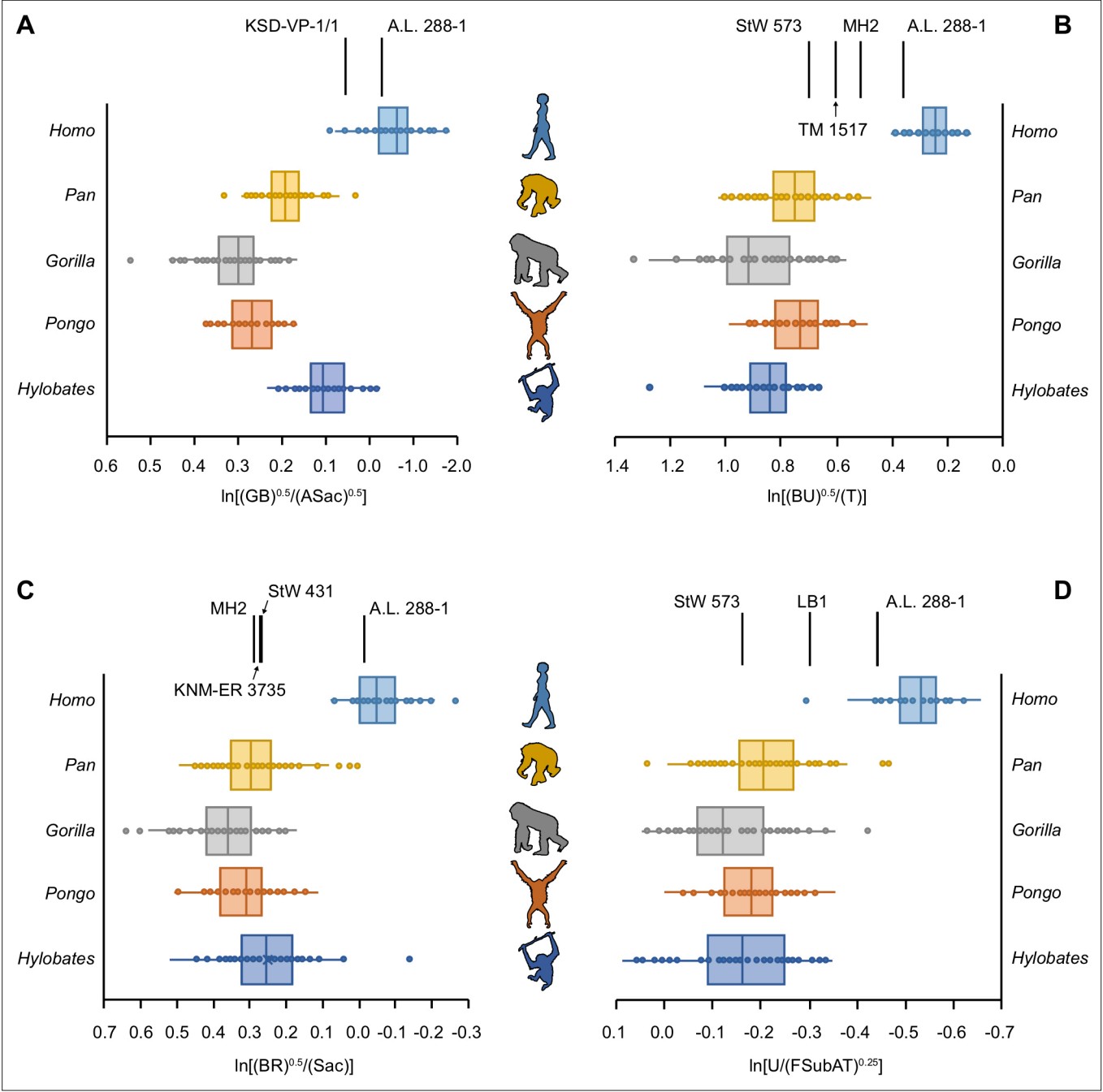

**Figure 2.** Relative limb size index (RLSI) in modern apes, humans, and fossil hominins. Notice that A.L. 288-1 (Lucy) falls within the modern human distribution for RLSI no matter which combination of upper to lower limb joint proportions is examined (**A–D**). (**A**) Human-like upper to lower limb joint proportions remain human-like on the basis of preserved elements in a second partial skeleton of *A. afarensis*, KSD-VP-1/1. However, all other partial skeletons of *Australopithecus* (**A–C**), *Paranthropus* (**B**) and early *Homo* (**C**) are more ape-like. A high, ape-like RLSI is present even in the late Pleistocene hominin *H. floresiensis* (**D**).

The online version of this article includes the following source data and figure supplement(s) for figure 2:

**Source data 1.** Raw measurements on extant primate skeletons.

**Figure supplement 1.** The relative limb size index of A.L. 288-1 falls within the interquartile range of modern humans and well outside the range of any modern ape.

**Figure supplement 2.** The relative limb size index of KSD-VP-1/1 falls within the interquartile range of modern humans and gibbons and is within the range of *Pan* when outliers are considered.

*Figure 2 continued on next page*

*Figure 2 continued*

**Figure supplement 3.** Here, the relative limb size index is a simple ratio of the glenoid size to the width of the talar body—measurements known from the juvenile *Australopithecus afarensis* skeleton from Dikika, Ethiopia (measurements published in *Green and Alemseged, 2012*; *DeSilva et al., 2018b*).

**Figure supplement 4.** The relative limb size index of KNM-WT 15000 is human-like and well outside the range of modern apes.

**Figure supplement 5.** Relative limb size index (RLSI) of LES 1 (Neo) from *Homo naledi*.

**Figure supplement 6.** The relative limb size index of StW 431 is decidedly ape-like, positioned within the interquartile range of hylobatids and within the range of all modern great apes, well outside the distribution of modern humans.

**Figure supplement 7.** The relative limb size index of MH1 is decidedly ape-like, positioned within the interquartile range of hylobatids and chimpanzees and within the range of gorillas and orangutans, well outside the distribution of modern humans.

**Figure supplement 8.** The relative limb size index of MH2 falls between the distribution in modern apes and modern humans, with the closest extant values being gibbons and chimpanzees.

**Figure supplement 9.** The relative limb size index of StW 573 falls between the distribution in modern apes and modern humans, with the closest extant value being a chimpanzee outlier.

**Figure supplement 10.** The relative limb size index of TM 1517 is ape-like, positioned within the range of gorillas, chimpanzees, and orangutans, well outside the distribution of modern humans.

**Figure supplement 11.** The relative limb size index of KNM-ER 1500 falls within the range of data in African apes and is outside the range of modern humans.

**Figure supplement 12.** The relative limb size index of OH 80 is chimpanzee-like, falling within the range of data in the other apes, and outside the range in modern humans.

**Figure supplement 13.** The relative limb size index of KNM-ER 1503/1504 is African ape-like, falling within the interquartile range of chimpanzees, within the range of gibbons, and outside the range in modern humans.

**Figure supplement 14.** The relative limb size index of BOU-VP-12/1 falls within the interquartile range of gorillas and within the full range of chimpanzees.

**Figure supplement 15.** The relative limb size index of KNM-ER 3735 is decidedly ape-like, falling with the range of distribution of chimpanzees, gorillas, orangutans, and gibbons, and well outside the range of modern humans.

**Figure supplement 16.** Relative limb size index of LB 1 (Flo) from *Homo floresiensis*.

**Figure supplement 17.** Limiting the comparison to a ratio between the radial head and the subtrochanteric region of the femur, only A.L. 288-1 falls within the interquartile range of *Homo sapiens*.

**Figure supplement 18.** Limiting the comparison to the biepicondylar breadth of humerus, radial head, ulna olecranon width, acetabulum height, and sacral size, only A.L. 288-1 falls within the interquartile range of *Homo sapiens*.

**Figure supplement 19.** Limiting the comparison to the biepicondylar breadth of humerus, ulna olecranon width, femoral head diameter, femoral subtrochanteric size, and sacral size, A.L. 288-1 and KNM-WT 15000 both fall within the human data range.

**Figure supplement 20.** Limiting the comparison to humeral head diameter, ulna olecranon width, femoral head diameter, femoral subtrochanteric size, and sacral size, A.L. 288-1 and LES 1 fall within the human distribution while KNM-WT 15000 is below the human data range.

**Figure supplement 21.** Limiting the comparison to glenoid size, humeral head diameter, biepicondylar width of humerus, radial head width, ulna olecranon width, femoral head diameter, talar trochlea width, and sacral size, A.L. 288-1 is barely within the human range while MH 2 is just below the range of extant apes.

**Figure supplement 22.** Limiting the comparison to biepicondylar width of humerus, femoral head diameter, and femoral subtrochanteric size, A.L. 288-1 and KNM-WT 15000 are within the human range.

**Figure supplement 23.** Limiting the comparison to glenoid size, humeral head diameter, biepicondylar width, ulna olecranon width, radial head, femoral head diameter, femoral subtrochanteric size, acetabulum height, and talar trochlea width, A.L. 288-1 is within the human range and StW 573 is at the bottom of the chimpanzee data range.

**Figure supplement 24.** Limiting the comparison to ulna olecranon width, radial head width, femoral subtrochanteric size, and talar trochlea width, A.L. 288-1 is within the human range, KNM-ER 1500 is in the bottom range of gorillas and chimpanzees, and StW 573 is within the interquartile range of the gorillas and chimpanzees.

**Figure supplement 25.** Limiting the comparison to glenoid size, humeral head diameter, biepicondylar width, ulna olecranon width, femoral head diameter, femoral subtrochanteric size, talar trochlea width, and sacral size, A.L. 288-1 is within the human range, while KNM-WT 15000 is just outside (below) the human range.

**Figure supplement 26.** Limiting the comparison to glenoid size and talar trochlea width, KNM-WT 15000 and DIK-1/1 are within the modern human range while A.L. 288-1 is just beyond the human data range, within the chimpanzee and gorilla range.

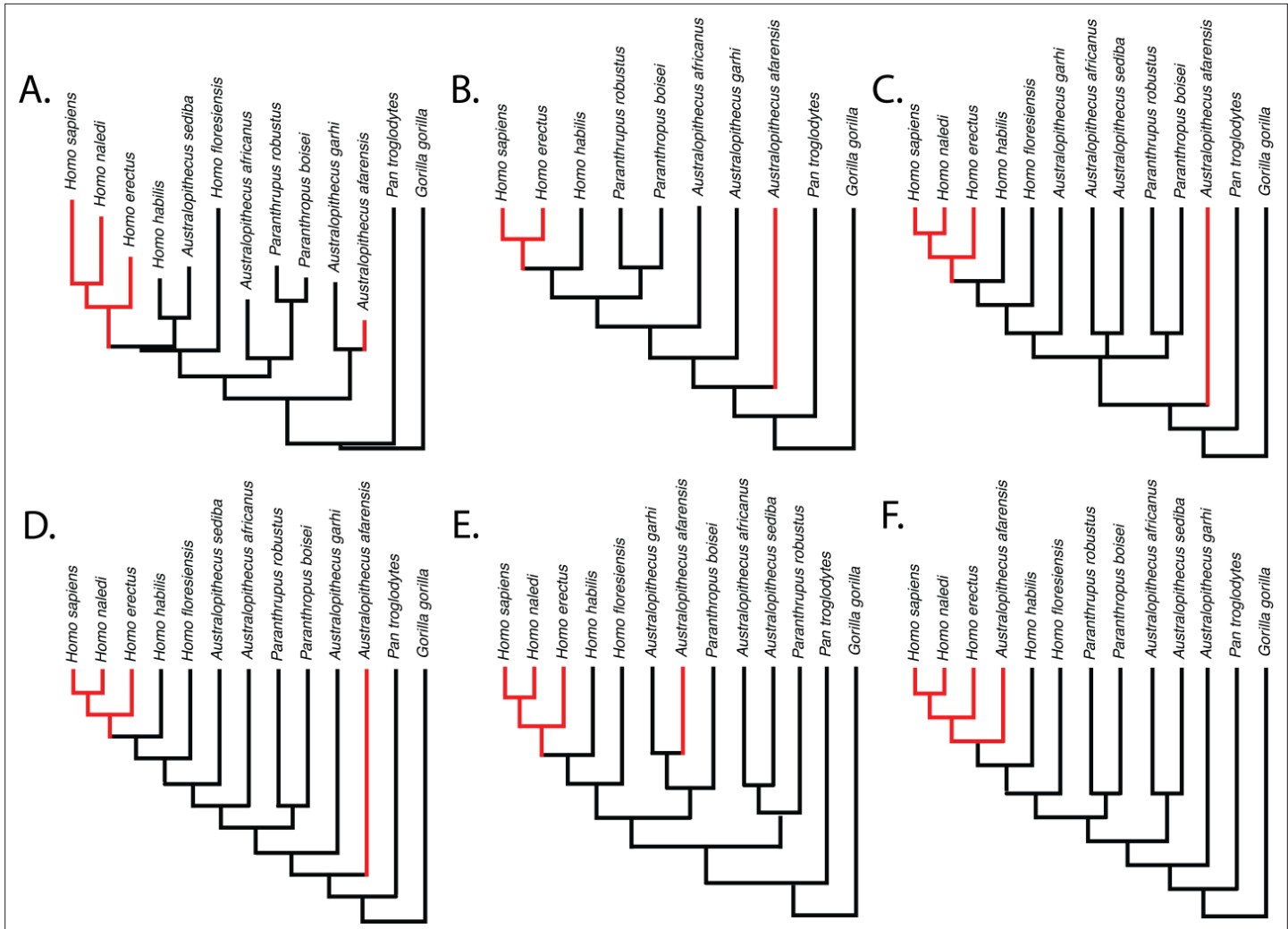

**Figure 3.** Relative limb size index (RLSI; high in black; low in red) for the taxa examined in this study. The phylogenies in (**A**) and (**B**) are from ***Dembo et al., 2016*** (**A**) and ***Mongle et al., 2019*** (**B**). The phylogenies in (**C–E**) presented above are informed by various hypotheses about the relationships of *Australopithecus* and *Paranthropus* taxa that have been published but not recovered in formal phylogenetic analyses. These include the hypothesis that *Australopithecus garhi* is a unique ancestor of *Homo* (**C**; ***Asfaw et al., 1999***), that *Australopithecus sediba* is a unique ancestor of *Homo* (**D**; ***Berger et al., 2010***; ***Irish et al., 2013***), and the hypothesis that *Paranthropus* is actually polyphyletic (**E**; topology based on hypothetical tree presented in ***Wood and Schroer, 2017***). A hypothetical phylogeny in which *Australopithecus afarensis* is more derived than two species of *Homo* as well as all other *Australopithecus* and *Paranthropus* species (such as shown in **F**) would need to be correct for the pattern of RLSI in hominins to be best explained as anything other than homoplasy between *A. afarensis* and some later Pleistocene *Homo*.

morphological pattern provides a framework for interpreting the functional and evolutionary implications of joint proportions in fossil hominins.

It is noteworthy, however, that the ape RLSI did not always align with degree of arboreality (hylobatid > *Pongo* > *Pan* > *Gorilla*). As reported elsewhere (***Gordon et al., 2020***), the preserved anatomical elements used to calculate RLSI can impact where a taxon is positioned along a locomotor continuum. The relatively narrow great ape sacrum, hypothesized to facilitate entrapment of the lumbar vertebrae and stiffen the lower back during climbing, further increases the RLSI of *Pan*, *Gorilla*, and *Pongo* relative to the hylobatids (***Figure 4—figure supplements 1–4***). Additionally, while RLSI calculations that included the radial head and ulnar trochlear width separate the ape species by locomotor mode, those that include the glenoid size and the biepicondylar breadth do not. In fact, a *post hoc* examination of limb joint scaling found that while great apes and humans exhibit isometric scaling of the glenoid and humeral biepicondylar breadth relative to femoral head diameter, the hylobatids scale with negative allometry (glenoid m = 0.75; biepicondylar breadth m = 0.76). It is likely that

the differing sizes of these apes and the functional demands on the limb joints in arboreal apes across this size range are driving some of the unexpected RLSI results reported here.

Our study provides a fresh perspective on alternative hypotheses for the evolution of limb joint proportions introduced by previous workers (*McHenry and Berger, 1998*; *Green et al., 2007*; *Haeusler and McHenry, 2007*). The reconstruction of patterns of hominin evolution relies on phylogeny, and, since the early adoption of cladistics, no quantitative analysis of hominin phylogeny has recovered a sister taxon relationship between *A. afarensis* and *Homo* (*Dembo et al., 2016*; *Strait et al., 2015*). The recovery of purported *Homo* fossils significantly predating the appearance of *A. africanus* and *A. sediba* may falsify hypotheses of exclusive ancestry and descent (*Du and Alemseged, 2019*). However, the consistent placement of *A. africanus* and *A. sediba* near *Homo* implies that they share a more recent common ancestor than do *Homo* and *A. afarensis*, despite their temporal and geographic distance (*Dembo et al., 2016*; *Berger et al., 2010*; *Irish et al., 2013*; *Pickering et al., 2011*). The inclusion of *Paranthropus* and early *Homo* fossils here helps alleviate the evolutionary implications of uncertainty surrounding the phylogenetic positions of *Australopithecus* species. The homology of

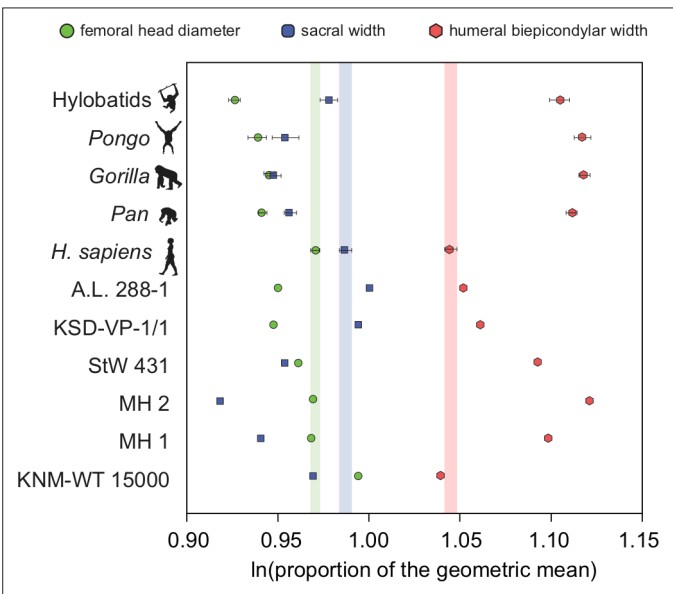

**Figure 4.** Additional evidence for homoplasy in relative limb size index (RLSI) between *A.afarensis* and *H. sapiens* is presented here. Only femoral head diameter, sacral width, and humeral biepicondylar width are considered in this analysis and all extant apes and hominin fossils are shown relative to the modern human condition (vertical-colored stripes). Horizontal bars are 95% confidence intervals (with those of human highlighted in vertical colored bars). Note that as in apes, *A. africanus* (StW 431) and *A. sediba* (MH1 and MH2) have a relatively large humeral biepicondylar width and relatively small sacrum. *A. afarensis* (A.L. 288-1 and KSD-VP-1/1) has a slightly larger biepicondylar breadth and sacral width with a slightly smaller femoral head relative to modern humans, though as already demonstrated, the overall RLSI is human-like. However, while *H. erectus* (KNM-WT 15000) also possesses a human-like RLSI, it is accomplished in a different anatomical manner. Notice that the colored dots (blue and green) are reversed in *H. erectus* relative to both *A. afarensis* and *H. sapiens,* meaning that in *H. erectus* the sacrum is smaller than expected (as in other australopiths) and the femoral head larger than expected. The BSN49/P27 *H. erectus* pelvis possesses a similarly small sacrum, indicating that this result is not solely a result of the juvenile status of KNM-WT 15000.

The online version of this article includes the following figure supplement(s) for figure 4:

**Figure supplement 1.** Same as for *Figure 2—figure supplement 1*, except in this relative limb size index (RLSI) analysis the sacrum has been removed.

**Figure supplement 2.** Same as for *Figure 2—figure supplement 2*, except in this relative limb size index (RLSI) analysis the sacrum has been removed.

**Figure supplement 3.** Same as for *Figure 2—figure supplement 4*, except in this relative limb size index (RLSI) analysis the sacrum has been removed.

**Figure supplement 4.** Same as for *Figure 2—figure supplement 5*, except in this relative limb size index (RLSI) analysis the sacrum has been removed.

a low RLSI in *A. afarensis* and later *Homo* can only be explained by an increasingly large number of evolutionary reversals. The independent evolution of similar limb joint proportions in *A. afarensis* and later *Homo* is a more parsimonious interpretation of the data.

There exists one additional piece of evidence that supports our interpretation that the low RLSI of *A. afarensis* and modern humans is homoplastic. Interestingly, the low RLSI of *A. afarensis* was achieved with a different morphological pattern compared to *H. erectus*. We found that *H. erectus* possessed a relatively smaller sacral body (like *A. africanus* and *A. sediba*) but a large femoral head relative to the upper limb than do modern humans; however, *A. afarensis* possessed a relatively small femoral head and large sacral body (*Figure 4*). This finding may further imply parallel evolution in limb joint proportions between *A. afarensis* and *H. sapiens*. The relatively small sacral body of the female *H. erectus* sacrum from Gona (*Simpson et al., 2008*, personal observation) demonstrates that these results are not the result of the juvenile status of KNM-WT 15000.

The morphology and functional anatomy of the axial skeleton, pelvis, and lower limb display unambiguous evidence for bipedal posture and locomotion in *Australopithecus* and later hominins. Furthermore, the presence of traits potentially signifying the importance of arboreality among fossil hominins does not necessarily imply reduced bipedal competency. However, the distributions of RLSI data (*Figure 2—figure supplements 17–26*), along with observations of other regional anatomies, imply differences among hominins in their adaptation to terrestrial, heel-striking bipedality. *A. afarensis* has relatively larger lower limb joints than any other early hominin currently known and possesses features of the foot and ankle that imply bipedal performance capabilities exceeding those of later early hominins. These traits include a more robust calcaneal tuber, a flatter subtalar joint, and a more plantarly oriented fourth metatarsal diaphysis and talonavicular joint (reviewed in *DeSilva et al., 2019*). The available morphological evidence suggests that, compared to other Plio-Pleistocene hominins, *A. afarensis* was better able to withstand the stresses and strains induced by the repetitive loading of the lower limb in frequent terrestrial bipedalism.

Over the past three decades, significant emphasis has been placed on the retention of ape-like characters in *Australopithecus* and *Paranthropus* since they could have been maintained through stabilizing selection if arboreality was a significant part of their positional repertoires (*Stern, 2000*). However, many researchers have repeatedly noted the difficulty of distinguishing the effects of stabilizing selection from those of evolutionary inertia (or 'lag'). This critique is rooted in a maximum likelihood, character-based cladistic framework, which implicitly excludes information about the evolutionary process (e.g., evolutionary rates as a function of time). In other words, primitive retentions have the same meaning across varying temporal ranges in a traditional cladistic framework. The presence of presumed primitive, ape-like features in late *Australopithecus* and early *Homo* c. ~2 Ma implies that evolutionary processes, whether neutral (i.e., genetic drift) or non-neutral (i.e., directional selection), had not yet substantially modified them over a 4- to 5 million-year period given current estimates of the *Pan-Homo* divergence date. Therefore, in our view, primitive retentions in *Australopithecus*, *Paranthropus*, and *Homo* can be meaningful when interpreted within the context of the evolutionary process.

Regardless of assumptions underlying evolutionary processes, primitive retentions in late *Australopithecus* and early *Homo* occur alongside indirect evidence for arboreal activity in their trabecular bone density patterns and long bone diaphyseal properties. A recent study of trabecular bone density of the non-pollical metacarpal heads of *A. sediba* showed a close morphometric affinity with extant orangutans, despite having human-like hand proportions (*Dunmore et al., 2020*). The external structure and internal trabecular morphology of the *A. sediba* hand are consistent with the use of forceful metacarpophalangeal joint flexion, which is a requisite of forelimb-dominated, below-branch locomotion (*Dunmore et al., 2020*). The purportedly more ape-like limb joint and length proportions of the OH 62 partial skeleton are supported by a more chimpanzee-like humeral cross-sectional geometry, implying that the *H. habilis* upper limb was heavily built (*Ruff, 2009*). The femoral head trabecular bone density pattern of StW 311 from Sterkfontein attributed either to *P. robustus* or *Homo* sp. implies the more habitual use of flexed hip postures, which occurs during climbing (*Georgiou et al., 2020*).

In light of the congruence between the functional interpretations derived from the external and internal morphology of hominin postcranial fossils, we consider the limb joint proportions data presented here, and in previous studies, to be a reliable indicator of adaptation to arboreal locomotion. The relatively larger hindlimb joints of *A. afarensis* and later *Homo* are consistent with a

more pronounced terrestrial component of their positional repertoires, whereas the relatively larger upper limb joints of most *Australopithecus*, *Paranthropus*, and early *Homo* individuals indicate a more pronounced arboreal component. The low RLSI of *A. afarensis* does not imply a lack of arboreal activity given the evidence it climbed trees (*Stern and Susman, 1983*; *Ruff et al., 2016*; *Green and Alemseged, 2012*; *DeSilva et al., 2018b*). The glenoid:talus proportions of the DIK-1-1 juvenile are human-like and distinct from this ratio in modern apes or in other species of *Australopithecus*, despite the presence of a *Gorilla*-like scapula (*Green and Alemseged, 2012*) and medial cuneiform (*DeSilva et al., 2018b*) indicating increased arboreal competency among juveniles. Furthermore, although *A. afarensis* had a more modern human-like RLSI, it possessed relatively longer arms and shorter legs than modern humans (*Holliday, 2012*), with more chimpanzee-like humeral-femoral strength proportions (*Ruff et al., 2016*), suggesting that the body plan and positional repertoire of *A. afarensis* were unique and unlike any living taxon.

We acknowledge that one of the limitations of our study includes uncertainty surrounding the taxonomic affinity of partial skeletons such as KNM-ER 1500. However, our evolutionary interpretation would not be altered by accepting the alternative interpretation of KNM-ER 1500 as *H. habilis*. Our finding that *Paranthropus* had a high RLSI is consistent with recent evidence suggesting the presence of heavily built, somewhat ape-like, distal humeri in this genus (*Lague et al., 2019*). Additionally, for some specimens (e.g. KNM-ER 1503/1504, TM 1517), uncertainty remains about whether they represent a single partial skeleton, but recent work supports the single-individual hypothesis for TM 1517 (*Cazenave et al., 2020*). Finally, two skeletons used in this study are juveniles (MH1 and KNM-WT 15000), though they are near skeletal maturation. Despite their juvenile status, MH1 has more ape-like limb joint proportions, whereas KNM-WT 15000 is more modern human-like. Fortunately, the limb joint proportions of *A. sediba* are represented by the MH2 adult specimen. A future study could evaluate the relative size and ontogenetic scaling of limb joint proportions across hominoids to evaluate the morphometric and functional affinities of juvenile hominin specimens in greater detail (e.g., DIK-1-1).

Despite these minor caveats, the pattern of limb joint proportions in hominins is clear. Partial skeletons belonging to *A. afarensis*, *H. erectus*, and *H. naledi* are human-like, with a low RLSI, whereas all others are more ape-like, with a high RLSI. These data strongly suggest that *A. afarensis* was a committed terrestrial biped that evolved adaptations to limit the larger lower limb stresses and strains characteristic of bipedal locomotion, as also occurred in later Pleistocene *Homo* (but not *H. floresiensis*). Other species of *Australopithecus*, *Paranthropus*, and early members of the genus *Homo* appear to have been less committed terrestrial bipeds that retained adaptations to the arboreal milieu. Overall, our analysis provides resolution on a long-standing hypothesis that *A. afarensis* evolved its low RLSI independently of some later Pleistocene hominins.

## Materials and methods

The comparative sample includes hylobatids (all four genera are represented, total N = 69; *Hoolock*: N = 11; *Hylobates*: N = 39; *Nomascus*: N = 5; *Symphalangus*: N = 14), *Pongo* spp. (total N = 50; *Pongo abelii*: N = 13; *Pongo pygmaeus*, N = 37), *Gorilla* spp. (total N = 131; *Gorilla beringei*: 55; *Gorilla gorilla*: N = 76), *Pan* spp. (total N = 124; *Pan paniscus*: N = 26, *Pan troglodytes*: N = 98), and *H. sapiens* (N = 67). We measured adult specimens from the Harvard Museum of Comparative Zoology, the American Museum of Natural History, and the Cleveland Museum of Natural History. Data for hominin partial skeletons (N = 16) were acquired from published literature and/or measured on original fossils using Mitutoyo calipers. In some cases, casts from the Dartmouth Paleoanthropology lab were used to confirm published measurements.

Seven measurements were taken at the shoulder and elbow joints to represent the upper body (*Figure 5*): scapular glenoid superoinferior (SI) height and maximum mediolateral (ML) width, humeral head SI height and anteroposterior (AP) width, humeral biepicondylar breadth, ulnar olecranon width, and radial head semimajor axis diameter. Seven measurements were taken at the hip, lumbosacral, and talocrural joints to represent the lower body (*Figure 5*): acetabulum SI height, SI femoral head diameter, femoral subtrochanteric ML width and AP breadth, sacral (S1) body maximum ML width and AP diameter at midline, and width of the talar trochlear apex taken at the midpoint. Mean scapular glenoid and humeral head joint size was calculated using a geometric mean of the SI and ML

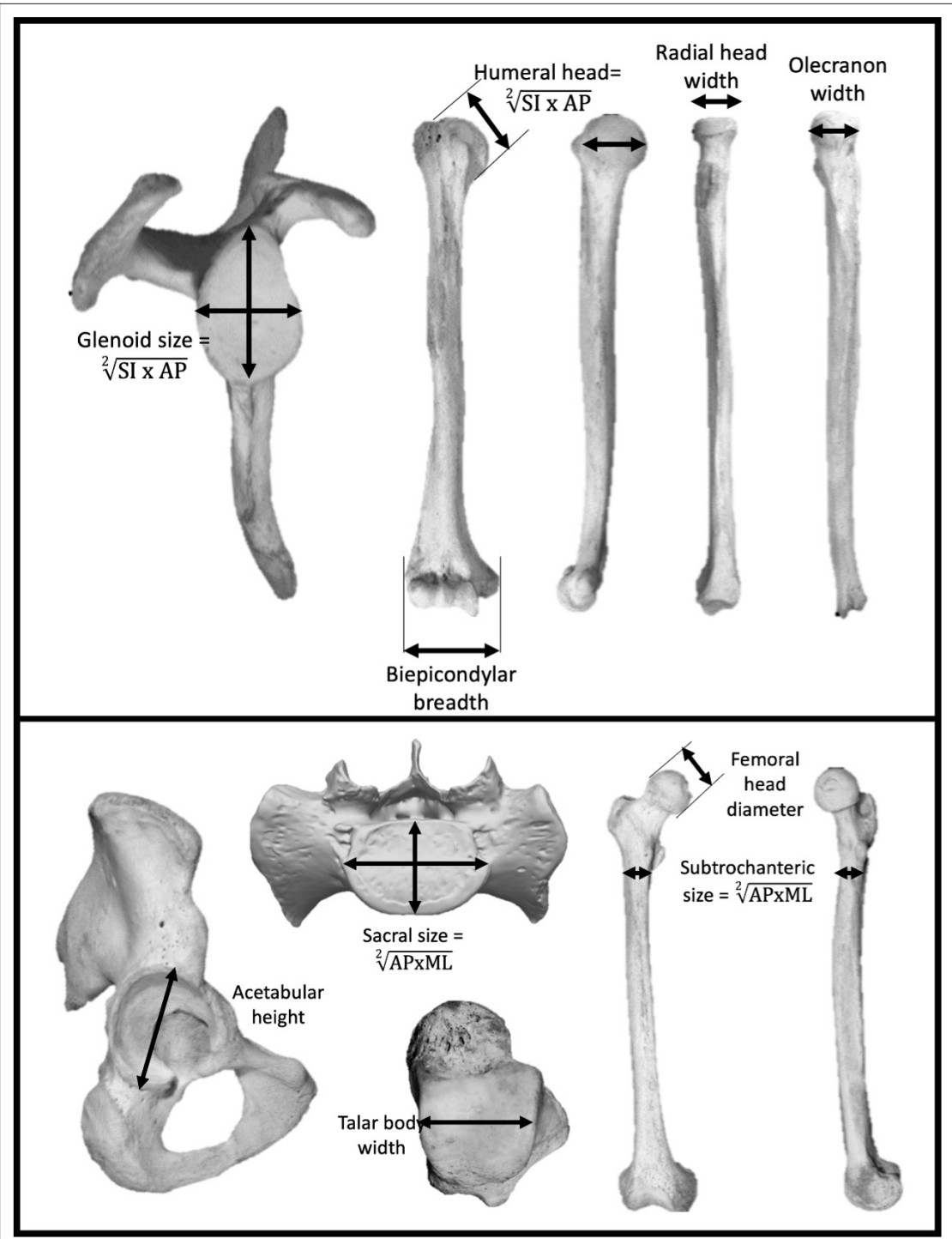

**Figure 5.** Linear measurements were taken on the upper limb (top) and lower limb (bottom). Limb joint proportions were calculated using the relative limb size index, which is the logged ratio of geometric means calculated from forelimb and hindlimb measurements shown above (*Green et al., 2007*).

dimensions. Mean femoral subtrochanteric and sacral body size was calculated using a geometric mean of the AP and ML dimensions.

We quantified limb joint proportions per individual using the RLSI (*Green et al., 2007*). The RLSI is the logged ratio of geometric means calculated from forelimb and hindlimb measurements (*Green et al., 2007*). Geometric means are typically used as size proxies over arithmetic means because they accommodate measurements with different ranges, which is common for morphometric measurements, and

therefore normalize the weight of individual measurements (*Jungers et al., 1995*). Logging the ratio is necessary because ratios of normally distributed data cannot be normally distributed and thus violate the assumptions of statistical tests (e.g., *Green et al., 2007*; *Smith, 1999*). The measurements used to calculate the limb joint proportions of hominin specimens varied depending on which measurements were preserved in the fossil (*Supplementary file 1*). Separate comparative analyses including different ratios were conducted for each hominin partial skeleton to maximize the fossil sample.

To visualize evolutionary scenarios, we conducted ancestral states using parsimony on a variety of phylogenetic hypotheses. These hypotheses included both formal cladistic analyses (*Dembo et al., 2016*; *Strait et al., 2015*; *Mongle et al., 2019*) and published hypotheses that have not been recovered in phylogenetic analyses (*Asfaw et al., 1999*; *Wood and Schroer, 2017*; *Villmoare, 2018*).

## Acknowledgements

The authors are grateful to the many individuals who made our data collection possible: Yohannes Haile-Selassie and Lyman Jellema (CMNH); Mark Omura (MCZ), Olivia Herschensohn, Kora Welsh, and Michèle Morgan (Harvard Peabody Museum of Archaeology and Ethnology); Neil Duncan, Eleanor Hoeger, Sara Ketelsen, Aja Marcato, Brian O'Toole, Marisa Surovy, and Eileen Westwig (AMNH); Emmanuel Gilissen and Wim Wendelen (Royal Museum for Central Africa); Lee Berger, Sifelani Jirah, and Bernhard Zipfel (Evolutionary Studies Institute, University of the Witwatersrand); Lazarus Kgasi, Stephany Potze, and Mirriam Tawane (Ditsong Museum of Natural History); Jared Assefa, Tomas Getachew, Getachew Senishaw, and Yonas Yilma (National Museum of Ethiopia, Authority for Research and Conservation of Cultural Heritage, and the Ethiopian Ministry of Culture and Tourism); Emma Mbua, Fredrick Manthi, and Job Kibii (National Museums of Kenya). We are additionally grateful to Matt Tocheri and Manuel Domínguez-Rodrigo for providing casts of hominin material included in this study.

## Additional information

### Funding

| Funder | Grant reference number | Author |
|---|---|---|
| Leakey Foundation | | Scott A Williams |
| Dartmouth College | | Jeremy M DeSilva |

The funders had no role in study design, data collection and interpretation, or the decision to submit the work for publication.

### Author contributions

Anjali M Prabhat, Conceptualization, Data curation, Formal analysis, Investigation, Methodology, Writing – original draft, Writing – review and editing; Catherine K Miller, Data curation, Formal analysis, Methodology, Writing – original draft; Thomas Cody Prang, Conceptualization, Data curation, Formal analysis, Methodology, Writing – review and editing; Jeffrey Spear, Data curation, Formal analysis, Methodology, Writing – review and editing; Scott A Williams, Conceptualization, Formal analysis, Methodology, Supervision, Writing – review and editing; Jeremy M DeSilva, Conceptualization, Formal analysis, Investigation, Methodology, Supervision, Writing – original draft, Writing – review and editing

### Author ORCIDs

Anjali M Prabhat (iD) https://orcid.org/0000-0003-0654-3455
Catherine K Miller (iD) https://orcid.org/0000-0002-0352-3777
Thomas Cody Prang (iD) http://orcid.org/0000-0003-3032-8309
Jeffrey Spear (iD) http://orcid.org/0000-0002-0290-7090
Scott A Williams (iD) https://orcid.org/0000-0001-7860-8962
Jeremy M DeSilva (iD) https://orcid.org/0000-0001-7010-1155

Decision letter and Author response
Decision letter https://doi.org/10.7554/eLife.65897.sa1
Author response https://doi.org/10.7554/eLife.65897.sa2

## Additional files

### Supplementary files

• Supplementary file 1. The table shows which joint measurements were taken for each fossil specimen as well as the formula used to express its relative upper limb to lower limb ratio as a geometric mean. G: glenoid size; H: humeral head diameter; B: humeral biepicondylar breadth; U: ulna olecranon width; R: radial head diameter; F: femoral head diameter; Sub: femoral subtrochanteric size; A: acetabulum height; T: talar width; Sac: sacral size. Cases in which only a single upper and/or lower limb measurement was possible (DIK-1-1, BOU-VP-12/1, OH 80, KNM-ER 1503/1504, KNM-ER 3735); the reported ratio does not include a geometric mean.

• Transparent reporting form

### Data availability

All data generated during this study are included in the manuscript (Table 1). Raw data from extant specimens appears as an Excel file in Figure 2-source data 1.

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
