## [Decision Letter]

**Acceptance summary:**

This paper will be useful for scholars interested in the evolution of bipedalism and the diversity of positional behavior in the hominin lineage and the pervasiveness of homoplasy/parallel evolution across various traits in human evolution. The authors expand the use of a proxy measure for arboreality/terrestriality to a much broader sample of fossil hominins than has previously been considered in a single analysis. Their results provide compelling support for the independent evolution of a high degree of terrestriality in one hominin species before it evolved in the modern human lineage.

**Decision letter after peer review:**

Thank you for submitting your article "Homoplasy in the evolution of modern human-like joint proportions in *Australopithecus afarensis*" for consideration by *eLife*. Your article has been reviewed by 2 peer reviewers, and the evaluation has been overseen by George Perry as the Senior and Reviewing Editor. The following individuals involved in review of your submission have agreed to reveal their identity: Adam Gordon (Reviewer #2); Kevin Hatala (Reviewer #3).

The reviewers were both broadly positive and also raised similar core points requiring your attention, for us to consider the manuscript for publication – please see the individual reviews below.

With respect to the overlapping point raised by both reviewers (concerning ancestral reconstruction and phylogenetic uncertainty) but with slightly different proposed solutions (removal of the ancestral reconstruction analysis in favor of a parsimony-based approach, versus incorporating a sensitivity analysis for the ancestral reconstruction): we offer you flexibility in ultimately determining how best to proceed, with our slight consensus preference for removing the ancestral reconstruction (as the results would still stand on their own without this).

*Reviewer #2:*

In this manuscript, the authors expand on earlier work examining articular and diaphyseal proportions in the fore- and hindlimb of extant hominoids and extinct hominins to address the question of homology or homoplasy for the previously-documented similarity of proportions in modern humans and *Australopithecus afarensis* in contrast to the more ape-like proportions found in other fossil hominins. The authors use logged ratios of measurements related to forelimb and hindlimb size following Green et al. (2007), including associated measurements from the glenoid, scapula, and sacrum. Overall the methods and sample are appropriate to the question at hand, the paper is well-organized and clearly written, and the conclusions follow logically from their results: *A. afarensis* appears to have evolved relatively large hindlimb articular and diaphyseal dimensions independently from later Homo, and in both cases these dimensions are likely to relate to a greater reliance on terrestrial bipedalism than in other hominins. That said, there are a couple of areas that would benefit from a bit more attention. In particular, ratio values for hylobatids that are unexpectedly intermediate between modern humans and other extant apes in some analyses need explaining. Because this pattern appears to be driven by relatively large sacral bodies in hylobatids, the authors should include an additional analysis to compare logged ratios among taxa when sacrum size is excluded from ratios. In addition, while their overall conclusion of homoplasy in *A. afarensis* and later *Homo* is well supported, it's important to note that the results of the ancestral reconstruction analyses are highly dependent on the selection of phylogeny and evolutionary model.

As one of the authors of Green et al. (2007) and a proponent of using these types of logged ratios to make inferences about positional behavior in extinct hominins, I'm pleased to see that the authors are using these logged ratios to examine a broader set of fossil taxa than they've been applied to before, and I think that the authors do a great job here. However, I also think that it's important to address the limitations of what can be inferred from these ratios and to highlight unusual patterns in the data.

For example, the rank-ordering of mean RLSI among extant ape taxa differs among the various ratios shown in Figure 2. I'd encourage the authors to take a look at our recent piece (Gordon et al., 2020) in which we consider how different logged limb proportions relate to various ways "degree of arboreality" can be defined. I'd like to see the authors discuss in a bit more detail just what can and cannot be inferred from comparisons of ratio values between fossils and extant taxa for these different ratios.

In particular, I think it's critically important that the authors spend more time discussing the unexpected pattern where Hylobates has the lowest (and thus more human-like) forelimb:hindlimb ratio of all of the extant apes in some cases, despite them using the most forelimb-dependent locomotor behavior of all of the taxa in this study. I think that the information highlighted in Figure 4 is really useful in partitioning out the relative contribution of different variables to denominator of these ratios, and that figure shows that the hylobatids have relatively large sacral bodies compared to other taxa, particularly in relation to femoral head size. And it's clear from Figure 2 that it's the ratios that include sacrum size in the denominator that produce lower values for the hylobatids than the other apes.

I'd argue that the authors need to point this pattern out, briefly comment on why hylobatids have relatively large sacral bodies compared to the other extant apes, and explain why this isn't (or is) a concern for interpreting fossil RLSI values that include sacrum size. Given that (1) inclusion of sacrum size shifts gibbon ratios towards modern humans and (2) sacrum size appears in the denominator of RLSI calculated for four of the five fossil specimens in the three taxa argued to be most human-like (i.e., A.L.288-1 and KSD-VP-1/1 in *A. afarensis*, KNM-WT 15000 in *H. erectus*, and LES 1 in *H. naledi*), it would be informative to recalculate RLSI without sacrum size in those specimens and the comparative sample to see if those fossil specimens still fall inside the human range and outside of the ape range. That's shown for a subset of Lucy's measurements in Figure 2B and D, but it should be shown for the most complete set of measurements (excluding sacrum size) for each of these four fossils. I expect the overall pattern will remain unchanged; I've calculated RLSI for these fossils (A.L.288-1, KSD-VP-1/1, KNM-WT 15000, and LES 1) and the extant sample means using the data reported in Table 1 and the RLSI equations reported in Table S1 (adjusted to exclude sacral size). In all cases the patterns appear to be consistent with the patterns when sacrum size is included, although without the data for individual comparative specimens I can't comment on the degree of overlap in the species ranges of the logged ratios. I'd suggest (1) adding these comparisons to the supplementary material as figures in the same style as the other supplementary figures, and (2) mentioning in the text that the exclusion of sacrum size tends to shift hylobatid logged ratios in line with the other apes but doesn't remove those fossil hominins from sitting in the modern human range.

Considering all of the above as well as the authors' discussion of the impact of taxonomic uncertainty in some cases, I believe that the authors present a compelling argument that joint proportions similar to those of modern humans are only found in *Homo erectus*, *H. naledi*, and *A. afarensis*, and notably are not found in other australopiths, *H. habilis*, or *H. erectus*. I also agree that even if one were to consider a number of different possible cladograms, the most parsimonious interpretation of these results is that *A. afarensis* developed low forelimb:hindlimb joint ratios independently from later *Homo* taxa. However, I don't think that the ancestral state reconstruction analysis adds much, if anything, to the argument. It's clear that ancestral reconstructions are highly dependent on branching topology, branch lengths, and assumptions regarding the underlying evolutionary model, and that errors in specifying any of these components can produce highly divergent results (e.g., see Ponti et al., 2020). Figure 3 essentially shows just what a parsimony-based analysis of the same data would show, with the addition of what I would argue are unrealistically precise estimates of ancestral values without any depiction of model-derived ranges of uncertainty around the ancestral estimates – which themselves are likely to underestimate the actual uncertainty due to probable misspecification of the phylogeny and/or evolutionary model. Personally, I am both a big proponent of phylogenetic methods for comparative analysis and pretty sour on ancestral reconstruction methods. So feel free to take my comments on this part of the manuscript with a grain of salt, but also with the understanding that I've been working with these methods and thinking about them for twenty years.

Comment on significance testing:

Some reviewers might take issue with the lack of p-values and/or confidence intervals for comparison of RLSI values in this manuscript, but I'm not one of them. The comparative samples for extant taxa are reasonably large, and the consistent differentiation of modern human ratios from extant apes combined with the placement of fossil hominin ratios within the range of one of those two groups but not the other is a more compelling argument to me than any p-values or confidence intervals would be. If the authors chose to calculate p-values or confidence intervals, they would undoubtedly support the arguments the authors make here, but I really don't think they're necessary.

References:

Gordon AD, Green DJ, Jungers WL, Richmond BG. 2020. Limb proportions and positional behavior: revisiting the theoretical and empirical underpinnings for locomotor reconstruction in *Australopithecus* africanus. In Zipfel B, Richmond BG, and Ward CV, eds.: Hominid Postcranial Remains from Sterkfontein, South Africa, 1936-1995. Advances in Human Evolution Series. Oxford University Press. pp. 321-334.

Ponti R, Arcones A, Vieetes DR. 2020. Challenges in estimating ancestral state reconstructions: the evolution of migration in Sylvia warblers as a study case. Integrative Zoology. 15:161-173.

*Reviewer #3:*

In this manuscript, Prabhat et al. evaluate limb joint proportions in modern humans and extant non-human apes, and through comparisons with these morphological patterns they infer locomotor behaviors from partial skeletons of various fossil hominins. By evaluating their results in the context of prior cladistic analyses, they draw further inferences about the evolutionary patterns that may be evident from fossil hominin limb joint proportions.

Studies of hominin limb joint size proportions have been conducted previously, but none have included such a broad sample of fossil taxa and none have focused to a similar degree on inferring evolutionary patterns from this aspect of postcranial morphology. The authors' analyses of joint proportions and their inferences of locomotor patterns are interesting, and they largely support results from other functional analyses of postcranial morphology. They find that skeletons of *Australopithecus afarensis*, *Homo erectus*, and *H. naledi* have joint size proportions that are similar to humans, and they infer that these taxa used a manner and/or degree of bipedalism that closely represented those of modern humans. Meanwhile, *A. africanus*, *A. sediba*, *Paranthropus robustus*, *P. boisei*, *H. habilis*, and *H. floresiensis* all show ape-like patterns of limb joint proportions, and the authors infer that their locomotor patterns would have included greater degrees of arboreality. More interesting, however, is that the authors proceed to interpret their results in the context of past cladistic analyses, and they find that the morphological pattern shared by *A. afarensis* and modern *H. sapiens* is likely the product of homoplasy rather than shared ancestry. This result suggests that the manner of bipedal locomotion used by *A. afarensis* was not a linear precursor to the form of bipedalism practiced by later species of the genus *Homo* and by modern humans today. I appreciate the authors' discussion of how this interpretative framework would change the ways in which we understand functional morphological patterns in *A. afarensis*, as they relate to bipedalism. For decades, paleoanthropologists have debated the "humanness" of their locomotor style. However, if we view their bipedalism as a reflection of homoplasy then it need not be connected in any way to the locomotion of modern humans and can instead be viewed simply as a different evolutionary solution. Overall, the analyses and results support the conclusions of these authors, and the results of this study are likely to have significant impacts on the field of paleoanthropology.

Perhaps the most important potential weaknesses, from my perspective, are related to (1) the dichotomization of human-like and ape-like morphologies, and (2) the degree to which the inference of evolutionary patterns hinges upon the particular phylogenetic tree selected for this analysis. I do not believe that either of these problems are entirely avoidable, but merely suggest that these are areas where the analyses and interpretations can be clarified and/or extended.

With respect to the first point, the dichotomous analysis means that non-human apes with different locomotor strategies and differing degrees of arboreality are grouped together, while humans stand as a sole extant contrast. The variations in joint size proportions among the non-human apes are not directly explored, so it remains difficult to know just how sensitive these variables are to different loading regimes (e.g., I would expect gibbons to experience the loading regimes most different from those of humans, yet in Figure 2 they fall closest to human-like joint size ratios in two of four panels and they are never the most different). Exploring this in greater depth could be very informative.

With respect to the second point, the differences between Figures 3A and 3B reflects sensitivity of the ancestral state reconstruction to the placement of *A. sediba*. While homoplasy is the most likely pattern in both scenario, it does appear that relatively subtle differences in the placement of this taxon influences the likelihood of ancestral states at various nodes and along certain branches throughout the tree. We can only work with the fossil record that we have, but I would be curious to see just how much difference it takes in order for a pattern other than homoplasy to emerge as most likely. Evaluating the robustness of this signal could even further strengthen the paper.

This manuscript was a pleasure to review. I found the questions interesting, the analyses appropriate, and the results and discussion compelling. It is also very well-written. I think this will make a significant contribution to the field, and it will prompt new lines of thought and inquiry as readers see these results and reflect on past analyses of joint size and limb length proportions in various fossil hominins. I really enjoyed thinking about what these results mean for how we interpret *A. afarensis* postcranial morphology and locomotion, which have certainly been subject to some of the most vigorous debates in paleoanthropology.

I believe there are just a few areas that could be clarified and/or explored to strengthen the manuscript even further. First, related to patterns among non-human apes, I found this fascinating and they seemed unexpected to me in some places (such as the gibbon example that I cited above). I wonder if you could explain in greater detail whether joint size proportions reflect the differences in locomotor behaviors between the non-human apes? I don't necessarily expect that a clear pattern will emerge, but it seems worth discussing since there is a lot of variation in the extent to which these various apes load their upper vs. lower limbs, not to mention other differences in body size, etc. If it is not purely loading regimes that affect these proportions (and I expect it is not that clear and straightforward) then it may be worth some page space (supplementary if there is no room in main text) to discuss what other factors may appear likely to influence these proportions.

Second, in my mind it could be useful to do some kind of sensitivity analysis related to the ancestral state reconstruction and subsequent evolutionary inferences. I know that deriving phylogenetic trees is not a goal of the current study, but it could be interesting and worthwhile to explore the robustness of the signal of homoplasy. Assuming that it is quite robust (which I expect to be the case), then it would further strengthen that component of the paper. If not, or if a certain taxon's place really disrupts the pattern, this would warrant some discussion.

Last, and this is not mentioned above, in the supplementary plots S2-S17 I think it would be very useful to display all possible fossil specimens in each analysis (e.g., use only the traits available from the specimen-of-focus, for as many specimens as possible). I'd be curious to know whether the ape-like or human-like signals hold up even when certain more complete specimens (e.g., A.L. 288-1, KNM-WT 15000), are "rarefied". On the other side of the equation, it may tell you whether signals from less complete specimens are likely to hold up if more of a given skeleton were to be found.

At a much more specific level, I thought the authors might consider re-phrasing lines 57-58, as the wording seems to imply that the loss of adaptations to arboreality and the transition to exclusive terrestrial bipedalism occurred in a linear pattern. In light of their past works, I don't think the authors would intend to make this implication. Similarly, line 316 discusses that *A. afarensis* and later Pleistocene *Homo* evolved similar bipedal adaptations "in parallel". To me, that seemed to imply parallel evolution, a term which is sometimes (though not always) defined as requiring overlap in time and/or place, which is clearly not the case here. I was certainly able to understand what you meant here, but I wonder if a term other than "parallel" may be better?

---

## [Author Response]

Reviewer #2:

In this manuscript, the authors expand on earlier work examining articular and diaphyseal proportions in the fore- and hindlimb of extant hominoids and extinct hominins to address the question of homology or homoplasy for the previously-documented similarity of proportions in modern humans and Australopithecus afarensis in contrast to the more ape-like proportions found in other fossil hominins. The authors use logged ratios of measurements related to forelimb and hindlimb size following Green et al. (2007), including associated measurements from the glenoid, scapula, and sacrum. Overall the methods and sample are appropriate to the question at hand, the paper is well-organized and clearly written, and the conclusions follow logically from their results: A. afarensis appears to have evolved relatively large hindlimb articular and diaphyseal dimensions independently from later Homo, and in both cases these dimensions are likely to relate to a greater reliance on terrestrial bipedalism than in other hominins. That said, there are a couple of areas that would benefit from a bit more attention. In particular, ratio values for hylobatids that are unexpectedly intermediate between modern humans and other extant apes in some analyses need explaining. Because this pattern appears to be driven by relatively large sacral bodies in hylobatids, the authors should include an additional analysis to compare logged ratios among taxa when sacrum size is excluded from ratios.

Thank you for this suggestion. We agree that the placement of the hylobatids in many of these analyses makes it challenging to interpret our findings along a terrestrial—arboreal continuum.

As proposed, we have re-run all of our analyses without contributions from the sacrum. Interestingly, while there is some shifting of the hylobatids, it is not enough to explain their intermediate positioning between the great apes and humans. We hypothesize, in part, an allometric explanation, as a result of further analyses which reveal that the biepicondylar width and glenoid size scale isometrically against femoral head diameter in all of the hominoids, except the hylobatids, which scale with negative allometry (m=0.75). Why this is the case remains unclear.

In addition, while their overall conclusion of homoplasy in A. afarensis and later Homo is well supported, it's important to note that the results of the ancestral reconstruction analyses are highly dependent on the selection of phylogeny and evolutionary model.

Yes, we agree. To that end, we have created a new Figure 3 in which six possible phylogenies are illustrated instead of the original two. Not only does this cover the different hypotheses current circulating, but also presents a phylogenetic scenario that would avoid RLSI homoplasy between later *Homo* and *A. afarensis*.

As one of the authors of Green et al. (2007) and a proponent of using these types of logged ratios to make inferences about positional behavior in extinct hominins, I'm pleased to see that the authors are using these logged ratios to examine a broader set of fossil taxa than they've been applied to before, and I think that the authors do a great job here. However, I also think that it's important to address the limitations of what can be inferred from these ratios and to highlight unusual patterns in the data.

For example, the rank-ordering of mean RLSI among extant ape taxa differs among the various ratios shown in Figure 2. I'd encourage the authors to take a look at our recent piece (Gordon et al., 2020) in which we consider how different logged limb proportions relate to various ways "degree of arboreality" can be defined. I'd like to see the authors discuss in a bit more detail just what can and cannot be inferred from comparisons of ratio values between fossils and extant taxa for these different ratios.

Thank you for alerting us to Gordon et al., 2020. We have added additional text that explores the limits of using RLSI to infer “degree of arboreality.”

In particular, I think it's critically important that the authors spend more time discussing the unexpected pattern where Hylobates has the lowest (and thus more human-like) forelimb:hindlimb ratio of all of the extant apes in some cases, despite them using the most forelimb-dependent locomotor behavior of all of the taxa in this study. I think that the information highlighted in Figure 4 is really useful in partitioning out the relative contribution of different variables to denominator of these ratios, and that figure shows that the hylobatids have relatively large sacral bodies compared to other taxa, particularly in relation to femoral head size. And it's clear from Figure 2 that it's the ratios that include sacrum size in the denominator that produce lower values for the hylobatids than the other apes.

I'd argue that the authors need to point this pattern out, briefly comment on why hylobatids have relatively large sacral bodies compared to the other extant apes, and explain why this isn't (or is) a concern for interpreting fossil RLSI values that include sacrum size. Given that (1) inclusion of sacrum size shifts gibbon ratios towards modern humans and (2) sacrum size appears in the denominator of RLSI calculated for four of the five fossil specimens in the three taxa argued to be most human-like (i.e., A.L.288-1 and KSD-VP-1/1 in A. afarensis, KNM-WT 15000 in H. erectus, and LES 1 in H. naledi), it would be informative to recalculate RLSI without sacrum size in those specimens and the comparative sample to see if those fossil specimens still fall inside the human range and outside of the ape range. That's shown for a subset of Lucy's measurements in Figure 2B and D, but it should be shown for the most complete set of measurements (excluding sacrum size) for each of these four fossils. I expect the overall pattern will remain unchanged; I've calculated RLSI for these fossils (A.L.288-1, KSD-VP-1/1, KNM-WT 15000, and LES 1) and the extant sample means using the data reported in Table 1 and the RLSI equations reported in Table S1 (adjusted to exclude sacral size). In all cases the patterns appear to be consistent with the patterns when sacrum size is included, although without the data for individual comparative specimens I can't comment on the degree of overlap in the species ranges of the logged ratios. I'd suggest (1) adding these comparisons to the supplementary material as figures in the same style as the other supplementary figures, and (2) mentioning in the text that the exclusion of sacrum size tends to shift hylobatid logged ratios in line with the other apes but doesn't remove those fossil hominins from sitting in the modern human range.

Thank you for this important observation. As suggested, we have rerun all of the analyses without the sacrum and include those as figure supplements. Additionally, we have added text to the main paper noting that the relatively large sacrum of gibbons is, in part, what is making their RLSI more human-like than the other hominoids.

Considering all of the above as well as the authors' discussion of the impact of taxonomic uncertainty in some cases, I believe that the authors present a compelling argument that joint proportions similar to those of modern humans are only found in Homo erectus, H. naledi, and A. afarensis, and notably are not found in other australopiths, H. habilis, or H. erectus. I also agree that even if one were to consider a number of different possible cladograms, the most parsimonious interpretation of these results is that A. afarensis developed low forelimb:hindlimb joint ratios independently from later Homo taxa. However, I don't think that the ancestral state reconstruction analysis adds much, if anything, to the argument. It's clear that ancestral reconstructions are highly dependent on branching topology, branch lengths, and assumptions regarding the underlying evolutionary model, and that errors in specifying any of these components can produce highly divergent results (e.g., see Ponti et al., 2020). Figure 3 essentially shows just what a parsimony-based analysis of the same data would show, with the addition of what I would argue are unrealistically precise estimates of ancestral values without any depiction of model-derived ranges of uncertainty around the ancestral estimates – which themselves are likely to underestimate the actual uncertainty due to probable misspecification of the phylogeny and/or evolutionary model. Personally, I am both a big proponent of phylogenetic methods for comparative analysis and pretty sour on ancestral reconstruction methods. So feel free to take my comments on this part of the manuscript with a grain of salt, but also with the understanding that I've been working with these methods and thinking about them for twenty years.

We agree and have modified Figure 3 so that it is no longer depicting ancestral state reconstructions but instead maps dichotomized RLSI (human-like or ape-like) on six different possible phylogenies.

Comment on significance testing:

Some reviewers might take issue with the lack of p-values and/or confidence intervals for comparison of RLSI values in this manuscript, but I'm not one of them. The comparative samples for extant taxa are reasonably large, and the consistent differentiation of modern human ratios from extant apes combined with the placement of fossil hominin ratios within the range of one of those two groups but not the other is a more compelling argument to me than any p-values or confidence intervals would be. If the authors chose to calculate p-values or confidence intervals, they would undoubtedly support the arguments the authors make here, but I really don't think they're necessary.

We appreciate that the reviewer does not find additional statistical analysis necessary.

References:

Gordon AD, Green DJ, Jungers WL, Richmond BG. 2020. Limb proportions and positional behavior: revisiting the theoretical and empirical underpinnings for locomotor reconstruction in Australopithecus africanus. In Zipfel B, Richmond BG, and Ward CV, eds.: Hominid Postcranial Remains from Sterkfontein, South Africa, 1936-1995. Advances in Human Evolution Series. Oxford University Press. pp. 321-334.

Ponti R, Arcones A, Vieetes DR. 2020. Challenges in estimating ancestral state reconstructions: the evolution of migration in Sylvia warblers as a study case. Integrative Zoology. 15:161-173.

We now cite Gordon et al., 2020 in our paper. Thank you.

Reviewer #3:

In this manuscript, Prabhat et al. evaluate limb joint proportions in modern humans and extant non-human apes, and through comparisons with these morphological patterns they infer locomotor behaviors from partial skeletons of various fossil hominins. By evaluating their results in the context of prior cladistic analyses, they draw further inferences about the evolutionary patterns that may be evident from fossil hominin limb joint proportions.

Studies of hominin limb joint size proportions have been conducted previously, but none have included such a broad sample of fossil taxa and none have focused to a similar degree on inferring evolutionary patterns from this aspect of postcranial morphology. The authors' analyses of joint proportions and their inferences of locomotor patterns are interesting, and they largely support results from other functional analyses of postcranial morphology. They find that skeletons of Australopithecus afarensis, Homo erectus, and H. naledi have joint size proportions that are similar to humans, and they infer that these taxa used a manner and/or degree of bipedalism that closely represented those of modern humans. Meanwhile, A. africanus, A. sediba, Paranthropus robustus, P. boisei, H. habilis, and H. floresiensis all show ape-like patterns of limb joint proportions, and the authors infer that their locomotor patterns would have included greater degrees of arboreality. More interesting, however, is that the authors proceed to interpret their results in the context of past cladistic analyses, and they find that the morphological pattern shared by A. afarensis and modern *H. sapiens *is likely the product of homoplasy rather than shared ancestry. This result suggests that the manner of bipedal locomotion used by A. afarensis was not a linear precursor to the form of bipedalism practiced by later species of the genus Homo and by modern humans today. I appreciate the authors' discussion of how this interpretative framework would change the ways in which we understand functional morphological patterns in A. afarensis, as they relate to bipedalism. For decades, paleoanthropologists have debated the "humanness" of their locomotor style. However, if we view their bipedalism as a reflection of homoplasy then it need not be connected in any way to the locomotion of modern humans and can instead be viewed simply as a different evolutionary solution. Overall, the analyses and results support the conclusions of these authors, and the results of this study are likely to have significant impacts on the field of paleoanthropology.

Thank you. We agree and hope that this paper adds a fresh, new perspective on terrestrial bipedalism in *A. afarensis*.

Perhaps the most important potential weaknesses, from my perspective, are related to (1) the dichotomization of human-like and ape-like morphologies, and (2) the degree to which the inference of evolutionary patterns hinges upon the particular phylogenetic tree selected for this analysis. I do not believe that either of these problems are entirely avoidable, but merely suggest that these are areas where the analyses and interpretations can be clarified and/or extended.

With respect to the first point, the dichotomous analysis means that non-human apes with different locomotor strategies and differing degrees of arboreality are grouped together, while humans stand as a sole extant contrast. The variations in joint size proportions among the non-human apes are not directly explored, so it remains difficult to know just how sensitive these variables are to different loading regimes (e.g., I would expect gibbons to experience the loading regimes most different from those of humans, yet in Figure 2 they fall closest to human-like joint size ratios in two of four panels and they are never the most different). Exploring this in greater depth could be very informative.

We agree that there are some fascinating (albeit unexpected) patterns in the non-human ape data. As recommended by Reviewer #2, we have re-run the analysis with the sacrum removed and included these new figures in the supplementary material. We have also added brief text acknowledging that RLSI does not track the degree of arboreality in living apes and hypothesize that allometry may help explain some of these findings. A closer examination of the patterns in the ape data is currently underway and is intended to be the subject of a follow-up manuscript.

With respect to the second point, the differences between Figures 3A and 3B reflects sensitivity of the ancestral state reconstruction to the placement of A. sediba. While homoplasy is the most likely pattern in both scenario, it does appear that relatively subtle differences in the placement of this taxon influences the likelihood of ancestral states at various nodes and along certain branches throughout the tree. We can only work with the fossil record that we have, but I would be curious to see just how much difference it takes in order for a pattern other than homoplasy to emerge as most likely. Evaluating the robustness of this signal could even further strengthen the paper.

We agree and have modified Figure 3 so that there are now six different phylogenetic hypotheses considered. Additionally, we no longer include ancestral state reconstructions.

This manuscript was a pleasure to review. I found the questions interesting, the analyses appropriate, and the results and discussion compelling. It is also very well-written. I think this will make a significant contribution to the field, and it will prompt new lines of thought and inquiry as readers see these results and reflect on past analyses of joint size and limb length proportions in various fossil hominins. I really enjoyed thinking about what these results mean for how we interpret A. afarensis postcranial morphology and locomotion, which have certainly been subject to some of the most vigorous debates in paleoanthropology.

I believe there are just a few areas that could be clarified and/or explored to strengthen the manuscript even further. First, related to patterns among non-human apes, I found this fascinating and they seemed unexpected to me in some places (such as the gibbon example that I cited above). I wonder if you could explain in greater detail whether joint size proportions reflect the differences in locomotor behaviors between the non-human apes? I don't necessarily expect that a clear pattern will emerge, but it seems worth discussing since there is a lot of variation in the extent to which these various apes load their upper vs. lower limbs, not to mention other differences in body size, etc. If it is not purely loading regimes that affect these proportions (and I expect it is not that clear and straightforward) then it may be worth some page space (supplementary if there is no room in main text) to discuss what other factors may appear likely to influence these proportions.

We, too, are struggling to understand the pattern in the data and do not have a satisfactory explanation. We have added some text hypothesizing that perhaps body size is driving some of the higher RLSI values in the great apes. Additionally, the sacrum appears to be contributing somewhat to the low RLSI value in gibbons. Other than that, we are somewhat baffled. However, we plan to expand our dataset to include cercopithecoids and Miocene hominoids and anticipate having a better explanation for these data in the near future.

Second, in my mind it could be useful to do some kind of sensitivity analysis related to the ancestral state reconstruction and subsequent evolutionary inferences. I know that deriving phylogenetic trees is not a goal of the current study, but it could be interesting and worthwhile to explore the robustness of the signal of homoplasy. Assuming that it is quite robust (which I expect to be the case), then it would further strengthen that component of the paper. If not, or if a certain taxon's place really disrupts the pattern, this would warrant some discussion.

Thank you for this suggestion. After consulting this comment and those made by Reviewer #2 and the editor, we decided to redraw Figure 3 using six different possible phylogenies. We hope this satisfies the concerns of the reviewer.

Last, and this is not mentioned above, in the supplementary plots S2-S17 I think it would be very useful to display all possible fossil specimens in each analysis (e.g., use only the traits available from the specimen-of-focus, for as many specimens as possible). I'd be curious to know whether the ape-like or human-like signals hold up even when certain more complete specimens (e.g., A.L. 288-1, KNM-WT 15000), are "rarefied". On the other side of the equation, it may tell you whether signals from less complete specimens are likely to hold up if more of a given skeleton were to be found.

This is a great point and we have now added all of these graphs as figure supplements. We have maintained the individual supplementary plots (Figure 2—figure supplements 1-16) and now have added these others (Figure 2—figure supplements 17-26)

At a much more specific level, I thought the authors might consider re-phrasing lines 57-58, as the wording seems to imply that the loss of adaptations to arboreality and the transition to exclusive terrestrial bipedalism occurred in a linear pattern. In light of their past works, I don't think the authors would intend to make this implication. Similarly, line 316 discusses that A. afarensis and later Pleistocene Homo evolved similar bipedal adaptations "in parallel". To me, that seemed to imply parallel evolution, a term which is sometimes (though not always) defined as requiring overlap in time and/or place, which is clearly not the case here. I was certainly able to understand what you meant here, but I wonder if a term other than "parallel" may be better?

We have added words and/or changed the language in both of these sections to avoid any misunderstandings.